# Image Captioning: Transforming Objects into Words

**Simao Herdade, Armin Kappeler, Kofi Boakye, Joao Soares**
Yahoo Research
San Francisco, CA, 94103
{sherdade,kaboakye,jvbsoares}@verizonmedia.com, akappeler@apple.com

## Abstract

Image captioning models typically follow an encoder-decoder architecture which uses abstract image feature vectors as input to the encoder. One of the most successful algorithms uses feature vectors extracted from the region proposals obtained from an object detector. In this work we introduce the Object Relation Transformer, that builds upon this approach by explicitly incorporating information about the spatial relationship between input detected objects through geometric attention. Quantitative and qualitative results demonstrate the importance of such geometric attention for image captioning, leading to improvements on all common captioning metrics on the MS-COCO dataset. Code is available at `https://github.com/yahoo/object_relation_transformer`.

## 1 Introduction

Image captioning—the task of providing a natural language description of the content within an image—lies at the intersection of computer vision and natural language processing. As both of these research areas are highly active and have experienced many recent advances, progress in image captioning has naturally followed suit. On the computer vision side, improved convolutional neural network and object detection architectures have contributed to improved image captioning systems. On the natural language processing side, more sophisticated sequential models, such as attention-based recurrent neural networks, have similarly resulted in more accurate caption generation.

Inspired by neural machine translation, most conventional image captioning systems utilize an encoder-decoder framework, in which an input image is encoded into an intermediate representation of the information contained within the image, and subsequently decoded into a descriptive text sequence. This encoding can consist of a single feature vector output of a CNN (as in [25]), or multiple visual features obtained from different regions within the image. In the latter case, the regions can be uniformly sampled (e.g., [26]), or guided by an object detector (e.g., [2]) which has been shown to yield improved performance.

While these detection based encoders represent the state-of-the art, at present they do not utilize information about the spatial relationships between the detected objects, such as relative position and size. This information can often be critical to understanding the content within an image, however, and is used by humans when reasoning about the physical world. Relative position, for example, can aid in distinguishing "a girl riding a horse" from "a girl standing beside a horse". Similarly, relative size can help differentiate between "a woman playing the guitar" and "a woman playing the ukelele". Incorporating spatial relationships has been shown to improve the performance of object detection itself, as demonstrated in [9]. Furthermore, in machine translation encoders, positional relationships are often encoded, in particular in the case of the Transformer [23], an attention-based encoder architecture. The use of relative positions and sizes of detected objects, then, should be of benefit to image captioning visual encoders as well, as evidenced in Figure 1.

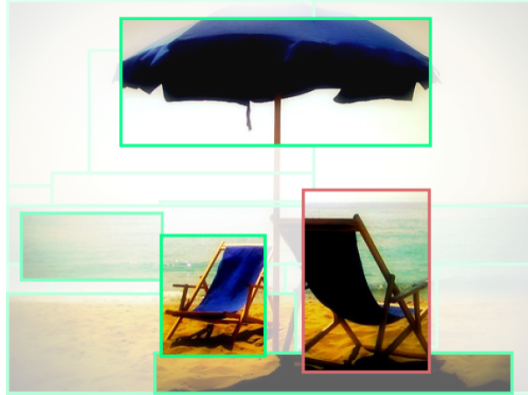

**Generated Caption**: *two beach chairs under an umbrella on the beach*

Figure 1: A visualization of self-attention in our proposed Object Relation Transformer. The transparency of the detected object and its bounding box is proportional to the attention weight with respect to the chair outlined in red. Our model strongly correlates this chair with the companion chair to the left, the beach beneath them, and the umbrella above them, relationships displayed in the generated caption.

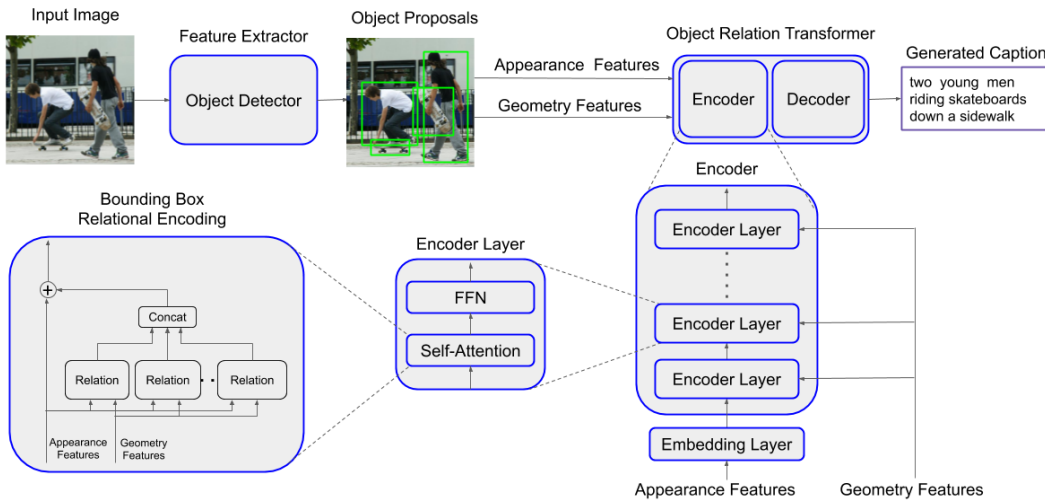

Figure 2: Overview of Object Relation Transformer architecture. The Bounding Box Relational Encoding diagram describes the changes made to the Transformer architecture

In this work, we propose and demonstrate the use of object spatial relationship modeling for image captioning, specifically within the Transformer encoder-decoder architecture. This is achieved by incorporating the object relation module of [9] within the Transformer encoder. The contributions of this paper are as follows:

- We introduce the Object Relation Transformer, an encoder-decoder architecture designed specifically for image captioning, that incorporates information about the spatial relationships between input detected objects through geometric attention.

- We quantitatively demonstrate the usefulness of geometric attention through both baseline comparison and an ablation study on the MS-COCO dataset.

- Lastly, we qualitatively show that geometric attention can result in improved captions that demonstrate enhanced spatial awareness.

# 2 Related Work

Many early neural models for image captioning [17, 12, 5, 25] encoded visual information using a single feature vector representing the image as a whole, and hence did not utilize information about objects and their spatial relationships. Karpathy and Fei-Fei in [11], as a notable exception to this global representation approach, extracted features from multiple image regions based on an R-CNN object detector [7] and generated separate captions for the regions. As a separate caption was generated for each region, however, the spatial relationship between the detected objects was not modeled. This is also true of their follow-on dense captioning work [10], which presented an end-to-end approach for obtaining captions relating to different regions within an image. Fang et al. in [6] generated image descriptions by first detecting words associated with different regions within the image. The spatial association was made by applying a fully convolutional neural network to the image and generating spatial response maps for the target words. Here again, the authors did not explicitly model any relationships between the spatial regions.

A family of attention based approaches [26, 30, 28] to image captioning have also been proposed that seek to ground the words in the predicted caption to regions in the image. As the visual attention is often derived from higher convolutional layers of a CNN, the spatial localization is limited and often not semantically meaningful. Most similar to our work, Anderson et al. in [2] addressed this limitation of typical attention models by combining a "bottom-up" attention model with a "top-down" LSTM. The bottom-up attention acts on mean-pooled convolutional features obtained from the proposed regions of interest of a Faster R-CNN object detector [20]. The top-down LSTM is a two-layer LSTM in which the first layer acts as a visual attention model that attends to the relevant detections for the current token and the second layers is a language LSTM that generates the next token. The authors demonstrated state-of-the-art performance for both visual question answering and image captioning using this approach, indicating the benefits of combining features derived from object detection with visual attention. Again, spatial information—which we propose in this work via geometric attention—was not utilized. Geometric attention was first introduced by Hu et al. for object detection in [9]. There, the authors used bounding box coordinates and sizes to infer the importance of the relationship of pairs of objects, the assumption being that if two bounding boxes are closer and more similar in size to each other, then their relationship is stronger.

The most successful subsequent work followed the above paradigm of obtaining image features with an object detector, and generating captions through an attention LSTM. As a way of adding global context, Yao et al. in [29] introduced two Graph Convolutional Networks: a semantic relationship graph, and a spatial relationship graph that classifies the relationship between two boxes into 11 classes, such as "inside", "cover", or "overlap". In contrast, our approach directly utilizes the size ratio and difference of the bounding box coordinates, implicitly encoding and generalizing the aforementioned relationships. Yang et al. in [27] similarly leveraged graph structures, extracting object image features into an image scene graph. In addition, they used a semantic scene graph (i.e., a graph of objects, their relationships, and their attributes) autoencoder on caption text to embed a language inductive bias in a dictionary that is shared with the image scene graph. While this model may learn typical spatial relationships found in text, it is inherently unable to capture the visual geometry specific to a given image. The use of self-critical reinforcement learning for sentence generation [21] has also proven to be important for state-of-the-art captioning approaches, such as those above. Liu et al. in [15] proposed an alternative reinforcement learning approach over a visual policy that, in effect, acts as an attention mechanism to combine features from the image regions provided by an object detector. The visual policy, however, does not utilize spatial information about these image regions.

Recent developments in NLP, namely the Transformer architecture [23] have led to significant performance improvements for various tasks such as translation [23], text generation [4], and language understanding [19]. In [22], the Transformer was applied to the task of image captioning. The authors explored extracting a single global image feature from the image as well as uniformly sampling features by dividing the image into 8x8 partitions. In the latter case, the feature vectors were fed in a sequence to the Transformer encoder. In this paper we propose to improve upon this uniform sampling by adopting the bottom-up approach of [2]. The Transformer architecture is particularly well suited as a bottom-up visual encoder for captioning since it does not have a notion of order for its inputs, unlike an RNN. It can, however, successfully model sequential data with the use of positional encoding, which we apply to the decoded tokens in the caption text. Rather than encode an order to

objects, our Object Relation Transformer seeks to encode how two objects are spatially related to each other and weight them accordingly.

## 3  Proposed Approach

Figure 2 shows an overview of the proposed image captioning algorithm. We first use an object detector to extract appearance and geometry features from all the detected objects in the image, as described in Section 3.1. Thereafter, we use the Object Relation Transformer to generate the caption text. Section 3.2 describes how we use the Transformer architecture [23] in general for image captioning. Section 3.3 explains our novel addition of box relational encoding to the encoder layer of the Transformer.

### 3.1  Object Detection

Following [2], we use Faster R-CNN [20] with ResNet-101 [8] as the base CNN for object detection and feature extraction. Using intermediate feature maps from the ResNet-101 as inputs, a Region Proposal Network (RPN) generates bounding boxes for object proposals. Using non-maximum suppression, overlapping bounding boxes with an intersection-over-union (IoU) exceeding a threshold of 0.7 are discarded. A region-of-interest (RoI) pooling layer is then used to convert all remaining bounding boxes to the same spatial size (e.g. $14 \times 14 \times 2048$). Additional CNN layers are applied to predict class labels and bounding box refinements for each box proposal. We further discard all bounding boxes where the class prediction probability is below a threshold of 0.2. Finally, we apply mean-pooling over the spatial dimension to generate a 2048-dimensional feature vector for each object bounding box. These feature vectors are then used as inputs to the Transformer model.

### 3.2  Standard Transformer Model

The Transformer [23] model consists of an encoder and a decoder, both of which are composed of a stack of layers (in our case 6). For image captioning, our architecture uses the feature vectors from the object detector as inputs and generates a sequence of words (i.e., the image caption) as outputs.

Every image feature vector is first processed through an input embedding layer, which consists of a fully-connected layer to reduce the dimension from 2048 to $d_{model} = 512$ followed by a ReLU and a dropout layer. The embedded feature vectors are then used as input tokens to the first encoder layer of the Transformer model. We denote $x_n$ as the $n$-th token of a set of $N$ tokens. For encoder layers 2 to 6, we use the output tokens of the previous encoder layer as the input to the current layer.

Each encoder layer consists of a multi-head self-attention layer followed by a small feed-forward neural network. The self-attention layer itself consists of 8 identical heads. Each attention head first calculates the queries $Q$, keys $K$ and values $V$ for the $N$ tokens as follows

$$Q = XW_Q, K = XW_K, V = XW_V, \tag{1}$$

where $X$ contains all the input vectors $x_1...x_N$ stacked into a matrix and $W_Q$, $W_K$, and $W_V$ are learned projection matrices.

The attention weights for the appearance features are then computed according to

$$\Omega_A = \frac{QK^T}{\sqrt{d_k}} \tag{2}$$

where $\Omega_A$ is an $N \times N$ attention weight matrix, whose elements $\omega_A^{mn}$ are the attention weights between the $m$-th and $n$-th token. Following the implementation of [23], we choose a constant scaling factor of $d_k = 64$, which is the dimension of the key, query, and value vectors. The output of the head is then calculated as

$$\text{head}(X) = \text{self-attention}(Q, K, V) = \text{softmax}(\Omega_A)V \tag{3}$$

Equations 1 to 3 are calculated for every head independently. The output of all 8 heads are then concatenated to one output vector and multiplied with a learned projection matrix $W_O$, i.e.,

$$\text{MultiHead}(Q, K, V) = \text{Concat}(\text{head}_1, \ldots, \text{head}_h)W_O \tag{4}$$

The next component of the encoder layer is the point-wise feed-forward network (FFN), which is applied to each output of the attention layer

$$\text{FFN}(x) = \max(0, xW_1 + b_1)W_2 + b_2 \tag{5}$$

where $W_1$,$b_1$ and $W_2$,$b_2$ are the weights and biases of two fully connected layers. In addition, skip-connections and layer-norm are applied to the outputs of the self-attention and the feed-forward layer.

The decoder then uses the generated tokens from the last encoder layer as input to generate the caption text. Since the dimensions of the output tokens of the Transformer encoder are identical to the tokens used in the original Transformer implementation, we make no modifications on the decoder side. We refer the reader to the original publication [23] for a detailed explanation of the decoder.

### 3.3  Object Relation Transformer

In our proposed model, we incorporate relative geometry by modifying the attention weight matrix $\Omega_A$ in Equation 2. We multiply the appearance based attention weights $\omega_A^{mn}$ of two objects $m$ and $n$, by a learned function of their relative position and size. We use the same function that was first introduced in [9] to improve the classification and non-maximum suppression stages of a Faster R-CNN object detector.

First we calculate a displacement vector $\lambda(m, n)$ for bounding boxes $m$ and $n$ from their geometry features $(x_m, y_m, w_m, h_m)$ and $(x_n, y_n, w_n, h_n)$ (center coordinates, widths, and heights) as

$$\lambda(m, n) = \left( \log \left( \frac{|x_m - x_n|}{w_m} \right), \log \left( \frac{|y_m - y_n|}{h_m} \right), \log \left( \frac{w_n}{w_m} \right), \log \left( \frac{h_n}{h_m} \right) \right), \tag{6}$$

The geometric attention weights are then calculated as

$$\omega_G^{mn} = ReLU \left( \text{Emb}(\lambda)W_G \right) \tag{7}$$

where $\text{Emb}(\cdot)$ calculates a high-dimensional embedding following the functions $PE_{pos}$ described in [23], where sinusoid functions are computed for each value of $\lambda(m, n)$. In addition, we multiply the embedding with the learned vector $W_G$ to project down to a scalar and apply the ReLU non-linearity. The geometric attention weights $\omega_G^{mn}$ are then incorporated into the attention mechanism according to

$$\omega^{mn} = \frac{\omega_G^{mn} \exp(\omega_A^{mn})}{\sum_{l=1}^{N} \omega_G^{ml} \exp(\omega_A^{ml})} \tag{8}$$

where $\omega_A^{mn}$ are the appearance based attention weights from Equation 2 and $\omega^{mn}$ are the new combined attention weights.

The output of the head can be calculated as

$$\text{head}(X) = \text{self-attention}(Q, K, V) = \Omega V \tag{9}$$

where $\Omega$ is the $N \times N$ matrix whose elements are given by $\omega^{mn}$.

The *Bounding Box Relational Encoding* diagram in Figure 2 shows the multi-head self-attention layer of the Object Relation Transformer. Equations 6 to 9 are represented with the *Relation* boxes.

## 4  Implementation Details

Our algorithm was developed in PyTorch using the image captioning implementation in [16] as our basis. We ran our experiments on NVIDIA Tesla V100 GPUs. Our best performing model was pre-trained for 30 epochs with a softmax cross-entropy loss using the ADAM optimizer with learning rate defined as in the original Transformer paper, with 20000 warmup steps, and a batch size of 10. We trained for an additional 30 epochs using self-critical reinforcement learning [21] optimizing for CIDEr-D score, and did early-stopping for best performance on the validation set (which contains 5000 images). On a single GPU the training with cross-entropy loss and the self-critical training take about 1 day and 3.5 days, respectively.

Table 1: Comparative analysis to existing state-of-the-art approaches. The model denoted as *Ours* refers to the Object Relation Transformer fine-tuned using self-critical training and generating captions using beam search with beam size 5.

| Algorithm | CIDEr-D | SPICE | BLEU-1 | BLEU-4 | METEOR | ROUGE-L |
|-----------|---------|-------|--------|--------|--------|---------|
| Att2all [21] | 114 | - | - | 34.2 | 26.7 | 55.7 |
| Up-Down [2] | 120.1 | 21.4 | 79.8 | 36.3 | 27.7 | 56.9 |
| Visual-policy[15] | 126.3 | 21.6 | – | 38.6 | 28.3 | 58.5 |
| GCN-LSTM [29][1] | 127.6 | 22.0 | 80.5 | 38.2 | 28.5 | 58.3 |
| SGAE [27] | 127.8 | 22.1 | **80.8** | 38.4 | 28.4 | **58.6** |
| Ours | **128.3** | **22.6** | 80.5 | **38.6** | **28.7** | 58.4 |

The models compared in sections 5.3-5.6 are evaluated after training for 30 epochs with standard cross-entropy loss, using ADAM optimization with the above learning rate schedule, and with batch size 15. The evaluation in those sections for the best performing models was obtained setting beam size to 2, in consistency with other research on image captioning optimization [21] (appendix A). Only in Table 1, for a fair comparison with other models in the literature, we present our result with the same beam size of 5 that other works have used to communicate their performance.

# 5 Experimental Evaluation

## 5.1 Dataset and Metrics

We trained and evaluated our algorithm on the Microsoft COCO (MS-COCO) 2014 Captions dataset [14]. We report results on the *Karpathy* validation and test splits [11], which are commonly used in other image captioning publications. The dataset contains 113K training images with 5 human annotated captions for each image. The *Karpathy* test and validation sets contain 5K images each. We evaluate our models using the CIDEr-D [24], SPICE [1], BLEU [18], METEOR [3], and ROUGE-L [13] metrics. While it has been shown experimentally that BLEU and ROUGE have lower correlation with human judgments than the other metrics [1, 24], the common practice in the image captioning literature is to report all the aforementioned metrics.

## 5.2 Comparative Analysis

We compare our proposed algorithm against the best results from a single model[1] of the self-critical sequence training (Att2all) [21] the Bottom-up Top-down (Up-Down) [2] baseline, and the three best to date image captioning models [15, 29, 27]. Table 1 shows the metrics for the test split as reported by the authors. Following the implementation of [2], we fine-tune our model using the self-critical training optimized for CIDEr-D score [21] and apply beam search with beam size 5, achieving a 6.8% relative improvement over the Up-Down baseline, as well as the state-of-the-art for the captioning specific metrics CIDEr-D, SPICE, as well as METEOR, and BLEU-4.

## 5.3 Positional Encoding

Our proposed geometric attention can be seen as a replacement for the positional encoding of the original Transformer network. While objects do not have an inherent notion of order, there do exist some simpler analogues to positional encoding, such as ordering by object size, or left-to-right or top-to-bottom based on bounding box coordinates. We provide a comparison between our geometric attention and these object orderings in Table 2. For box size, we simply calculate the area of each bounding box and order from largest to smallest. For left-to-right we order bounding boxes according to the x-coordinate of their centroids. Analogous ordering is performed for top-to-bottom using the centroid y-coordinate. Based on the CIDEr-D scores shown, adding such an artificial ordering to the detected objects decreases the performance. We observed similar decreases in performance across all other metrics (SPICE, BLEU, METEOR and ROUGE-L).

Table 2: Positional Encoding Comparison (models trained with softmax cross-entropy for 30 epochs)

| Positional Encoding | CIDEr-D |
|---|---|
| no encoding | 111.0 |
| positional encoding (ordered by box size) | 108.7 |
| positional encoding (ordered left-to-right) | 110.2 |
| positional encoding (ordered top-to-bottom) | 109.1 |
| geometric attention | 112.6 |

Table 3: Ablation Study. All metrics are reported for the validation and the test split, after training with softmax cross-entropy for 30 epochs. The Transformer (Transf) and the Object Relational Transformer (ObjRel Transf) is described in detail in Section 3

| Algorithm | | CIDEr-D | SPICE | BLEU-1 | BLEU-4 | METEOR | ROUGE-L |
|---|---|---|---|---|---|---|---|
| Up-Down + LSTM | val | 105.6 | 19.7 | 75.5 | 32.9 | 26.5 | 55.6 |
| | test | 106.6 | 19.9 | 75.6 | 32.9 | 26.5 | 55.4 |
| Up-Down + Transf | val | 110.5 | 20.8 | 75.2 | 33.3 | 27.6 | 55.8 |
| | test | 111.0 | 20.9 | 75.0 | 32.8 | 27.5 | 55.6 |
| Up-Down + ObjRel Transf | val | 113.2 | 21.0 | 76.1 | 34.4 | 27.7 | 56.4 |
| | test | 112.6 | 20.8 | 75.6 | 33.5 | 27.6 | 56.0 |
| Up-Down + ObjRel Transf | val | 114.7 | 21.1 | 76.5 | 35.5 | 27.9 | 56.6 |
| + Beamsize 2 | test | 115.4 | 21.2 | 76.6 | 35.5 | 28.0 | 56.6 |

## 5.4 Ablation Study

Table 3 shows the results for our ablation study. We show the Bottom-Up and Top-Down algorithm [2] as our baseline algorithm. The second row replaces the LSTM with a Transformer network. The third row includes the proposed geometric attention. The last row includes beam search with beam size 2. The contribution of the Object Relation Transformer is small for METEOR, but significant for CIDEr-D and the BLEU metrics. Overall we can see the most improvements on the CIDEr-D and BLEU-4 scores.

## 5.5 Geometric Improvement

In order to demonstrate the advantages of the geometric attention layer, we performed a more detailed comparison of the Object Relation Transformer against the Standard Transformer. For each of the considered metrics, we performed a two-tailed t-test with paired samples in order to determine whether the difference caused by adding the geometric attention was statistically significant. The metrics were first computed for each individual image in the test set for each of the two Transformer models, so that we are able to run the paired tests. In addition to the standard evaluation metrics, we also report metrics obtained from SPICE by splitting up the tuples of the scene graphs according to different semantic subcategories. For each subcategory, we are able to compute precision, recall, and F-scores. The measures we report are the F-scores computed by taking only the tuples in each subcategory. More specifically, we report SPICE scores for: Object, Relation, Attribute, Color, Count, and Size [1]. Note that for a given image, not all SPICE subcategory scores might be available. For example, if the reference captions for a given image have no mention of color, then the SPICE Color score is not defined and therefore we omit that image from that particular analysis. In spite of this, each subcategory analyzed had at least 1000 samples. For this experiment, we did not use self-critical training for either Transformer and they were both run with a beam size of 2.

The metrics computed over the 5000 images of the test set are shown in Tables 4 and 5. We first note that for all of the metrics, the Object Relation Transformer presents higher scores than the Standard Transformer. The score difference was statistically significant (using a significance level $\alpha = 0.05$) for CIDEr-D, BLEU-1, ROUGE-L (Table 4), Relation, and Count (Table 5). The significant improvements in CIDEr-D and Relation are in line with our expectation that adding the geometric attention layer would help the model in determining the correct relationships between objects. In addition, it is interesting to see a significant improvement in the Count subcategory of SPICE, from 11.30 to 17.51. Though image captioning methods in general show a large deficit in Count scores when compared with humans [1], we are able to show a significant improvement by adding explicit positional information. Some examples illustrating these improvements are presented in Section 5.6.

Table 4: Comparison of different captioning metrics for the Standard Transformer and our proposed Object Relation Transformer (denoted *Ours* below), trained with softmax cross-entropy for 30 epochs. The table shows that the Object Relation Transformer has significantly higher CIDEr-D, BLEU-1 and ROUGE-L scores. The p-values come from two-tailed t-tests using paired samples. Values marked in bold were considered significant at $\alpha = 0.05$.

| Algorithm | CIDEr-D | SPICE | BLEU-1 | BLEU-4 | METEOR | ROUGE-L |
|---|---|---|---|---|---|---|
| Standard Transformer | 113.21 | 21.04 | 75.60 | 34.58 | 27.79 | 56.02 |
| Ours | 115.37 | 21.24 | 76.63 | 35.49 | 27.98 | 56.58 |
| p-value | **0.01** | 0.15 | **<0.001** | 0.051 | 0.24 | **0.01** |

Table 5: Breakdown of SPICE metrics for the Standard Transformer and our proposed Object Relation Transformer (denoted *Ours* below), trained with softmax cross-entropy for 30 epochs. The table shows that the Object Relation Transformer has significantly higher Relation and Count scores. The p-values come from two-tailed t-tests using paired samples. Values marked in bold were considered significant at $\alpha = 0.05$.

| Algorithm | SPICE | | | | | | |
|---|---|---|---|---|---|---|---|
| | All | Object | Relation | Attribute | Color | Count | Size |
| Standard Transformer | 21.04 | 37.83 | 5.88 | 11.31 | 14.88 | 11.30 | 5.82 |
| Ours | 21.24 | 37.92 | 6.31 | 11.37 | 15.49 | 17.51 | 6.38 |
| p-value | 0.15 | 0.64 | **0.01** | 0.81 | 0.35 | **<0.001** | 0.34 |

## 5.6 Qualitative Analysis

To illustrate the advantages of the Object Relation Transformer relative to the Standard Transformer, we present example images with the corresponding captions generated by each model. The captions presented were generated using the following setup: both the Object Relation Transformer and the Standard Transformer were trained without self-critical training and both were run with a beam size of 2 on the 5000 images of the test set. We chose examples for which there were was a marked improvement in the score of the Object Relation Transformer relative to the Standard Transformer. This was done for the Relation and Count subcategories of SPICE scores. The example images and captions are presented in Tables 6 and 7. The images in Table 6 illustrate an improvement in determining when a relationship between objects should be expressed, as well as in determining what that relationship should be. An example of correctly determining that a relationship should exist is shown in the third image of Table 6, where the two chairs are actually related to the umbrella by being underneath it. In addition, an example where the Object Relation Transformer correctly infers the type of relationship between objects is shown in the first image of Table 6, where the man in fact is not *on* the motorcycle, but is *working on* it. The examples in Table 7 specifically illustrate the Object Relation Transformer's marked ability to better count objects.

Table 6: Example images and captions for which the SPICE **Relation** metric for Object Relation Transformer shows an improvement over the metric for the Standard Transformer.

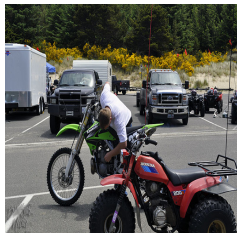   

| **Standard:** a man on a motorcycle on the road | a couple of bears standing on top of a rock | two chairs and an umbrella on a beach | a laptop computer sitting on top of a wooden desk |
|---|---|---|---|
| **Ours:** a man is working on a motorcycle in a parking lot | two brown bears standing next to each other on a rock | two beach chairs under an umbrella on the beach | a desk with a laptop and a keyboard |

Table 7: Example images and captions for which the SPICE **Count** metric for the Object Relation Transformer shows an improvement over the metric for the Standard Transformer.

| 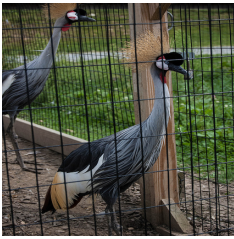 |  |  |  |
|---|---|---|---|
| **Standard:** a large bird is standing in a cage | a little girl sitting on top of a giraffe | a group of young men riding skateboards down a sidewalk | three children are sitting on a bunk bed |
| **Ours:** two large birds standing in a fenced in area | a giraffe with two kids sitting on it | two young men riding skateboards down a sidewalk | two young children are sitting on the bunk beds |

In order to better understand the failure modes of our model, we manually reviewed a set of generated captions. We used our best performing model—the Object Relation Transformer trained with self-critical reinforcement learning—with a beam size of 5 to generate captions for 100 randomly sampled images from the MS-COCO's test set. For each generated caption, we described the errors and then grouped them into distinct failure modes. An error was counted each time a term was wrong, extraneous, or missing. All errors were then tallied up, with each image being able to contribute with multiple errors. There were a total of 62 observed errors, which were grouped into 4 categories: 58% of the errors pertained to objects or things, 21% to relations, 16% to attributes, and 5% to syntax. Note that while these failure modes are very similar to the semantic subcategories from SPICE, we were not explicitly aiming to adhere to those. In addition, one general pattern that stood out were the errors in identifying rare or unusual objects. Some examples of unusual objects that were not correctly identified include: parking meter, clothing mannequin, umbrella hat, tractor, and masking tape. This issue is also noticeable, even if to a lesser degree, in rare relations and attributes. Another interesting observation was that the generated captions tend to be less descriptive and less discursive than the ground truth captions. The above results and observations can be used to help prioritize future efforts in image captioning.

## 6  Conclusion

We have presented the Object Relation Transformer, a modification of the conventional Transformer, specifically adapted to the task of image captioning. The proposed Transformer encodes 2D position and size relationships between detected objects in images, building upon the bottom-up and top-down image captioning approach. Our results on the MS-COCO dataset demonstrate that the Transformer does indeed benefit from incorporating spatial relationship information, most evidently when comparing the relevant sub-metrics of the SPICE captioning metric. We have also presented qualitative examples of how incorporating this information can yield captioning results demonstrating better spatial awareness.

At present, our model only takes into account geometric information in the encoder phase. As a next step, we intend to incorporate geometric attention in our decoder cross-attention layers between objects and words. We aim to do this by explicitly associating decoded words with object bounding boxes. This should lead to additional performance gains as well as improved interpretability of the model.

**Acknowledgments**

The authors would like to thank Ruotian Luo for making his image captioning code available on GitHub [16].

## Footnotes

[1]Some publications include results obtained from an ensemble of models. Specifically, the ensemble of two distinct graph convolution networks in GCN-LSTM [29] achieves a superior CIDEr-D score to our stand-alone model.

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
