[Reviews · NeurIPS 2019]

Reviewer 1



Summary - The proposed approach to image captioning extends two prior works, object-based Up-Down method of [2] and Transformer of [22] (already used for image captioning in [21]). Specifically, the authors integrate spatial relations between objects in the captioning Transformer model, proposing the Object Relation Transformer. The modification amounts to introducing an object relation module [9] into the encoding layer of the Transformer model. - The authors evaluate the proposed approach on the Karpathy split of MSCOCO dataset, demonstrating state-of-the-art performance on automatic metrics, in particular in CIDEr-D score. (The compared approaches only include Att2All [20] and Up-Down [2].) - A few ablations and baselines are present, including comparison to a standard Transformer model. Tests of statistical significance show that the proposed model outperforms the standard Transformer in terms of CIDEr-D, BLEU-1 and ROUGE-L, while SPICE-attribute breakdown shows improvement for Relation and Count categories. Qualitative results include examples where Object Relation Transformer leads to more correct spatial Relation and Count predictions. Originality - The proposed approach combines ideas from [2,9,22] in a straightforward manner. - Several prior works are not cited [A,B,C,D]. [A] proposed context-aware visual policy network, significantly improving over Up-Down. [B] introduced a GCN-LSTM model which integrates semantic and spatial object relationships. [C] further made use of scene graphs to integrate object, attribute, and relationship knowledge. Finally, recently [D] proposed “relational captioning”, a new task where multiple captions need to be generated for an image to summarize different relationships present in the image. Comparison to these works, both conceptual and empirical is necessary. [A] D. Liu, Z.-J. Zha, H. Zhang, Y. Zhang, and F. Wu. Context-aware visual policy network for sequence-level image captioning. In 2018 ACM Multimedia Conference on Multimedia Conference, pages 1416–1424. ACM, 2018. [B] T. Yao, Y. Pan, Y. Li, and T. Mei. Exploring visual relationship for image captioning. In Computer Vision–ECCV 2018, pages 711–727. Springer, 2018. [C] Yang, Xu, et al. "Auto-encoding scene graphs for image captioning." Proceedings of the IEEE Conference on Computer Vision and Pattern Recognition. 2019. [D] Kim, Dong-Jin, et al. "Dense relational captioning: Triple-stream networks for relationship-based captioning." Proceedings of the IEEE Conference on Computer Vision and Pattern Recognition. 2019. Quality - The proposed approach appears technically sound, as it extends prior work in a straightforward manner. - The very similar prior work [21], which also utilized Transformer for image captioning, is cited, but no empirical comparison is included. Also, recent results from other works are not included [A,B,C]. - Although comparison to state-of-the-art is incomplete (see above), the authors provide an informative ablation/baseline study, including tests of statistical significance, etc. - No evaluation on the online MSCOCO test server is included. The online test set is meant to demonstrates models’ performance in a blind scenario, where overfitting is less likely. - No human evaluation is included, although the authors are aware that some of the automatic metrics poorly reflect human preference (L178); in fact, all automatic metrics fall short of capturing human preference, thus human evaluation is desirable. - It would have been interesting to see some failure cases in additional to the presented (cherry-picked) success cases, or to see some discussion on that. Clarity - The paper is clearly written and was mostly easy to follow. - I do not find Figure 1 helpful in terms of illustrating the proposed approach, e.g. no explicit visualization of spatial relationships is present. The main idea is not obvious from looking at this figure. - It is somewhat confusing that different tables reflect different settings, e.g. with/without self-critical training, with beam size 5, 2 or 1 (?), and it is not always clear which case is shown and how different numbers in different tables relate to each other. - It would also be helpful to report the two standard settings, with beam size 5 with/without self-critical training, compared side-by-side, as in [2]. Significance - The idea of harnessing spatial relations for image captioning is not new [B,C]. The proposed approach is not particularly technically innovative, but appears quite effective. The overall improvement over prior work [2] is impressive, but in comparison to more recent works [A,B,C], the results are on par / only marginally better. Thus, I rate the significance of the presented contributions as Medium/Low. UPDATE The authors did not include empirical comparison to prior work [A,B,C] or human evaluation, promising to do so in the final version. The results on the test server are not put in context with any of the baselines. With neither a clear comparison to state-of-the-art [A,B,C], nor human evaluation (which is always desirable for tasks like captioning), I am not convinced about the overall significance/impact of the proposed method.

Reviewer 2



1) Originality: The authors propose an incremental modification to the transformer network which accounts for spatial relationship between the objects. The ideas of using transformer encoder+decoder architecture, pre-training the network with cross-entropy loss and fine-tuning in self-critical setup using CIDER scores with rollout as reward are borrowed from previous literature. 2) Quality: The proposed modification i.e. the geometric attention weights are supposed to encode geometric relationships between objects. The accuracy or success of this modification is measured via a proxy of better performance on captioning metrics and spice sub-scores for relation/count/size etc. Though the captioning metrics show improved performance and some qualitative results indicate that the network understands spatial relationships, it is hard to tell if the network is learning meaningful spatial relationships in its geometric attention weights. A visualization for the attention weights for an appropriate layer (say for the qualitative examples in Tab. 7) may be useful to demonstrate that the network does indeed learn spatial relationship information. Also, the method does not get a statistically significant improvement for the "size" sub-score of SPICE metric in Tab. 5. The proposed geometric attention weights use relative size of bounding boxes in eq. 6, but this does not translate into performance gain. 3) Clarity: The paper is well written and easy to follow. The contributions are stated clearly and sufficient details of the method are present. The experimental evaluations are properly described with sufficient details. 4) Significance: The paper proposes an incremental modification to the transformer networks for image captioning, which can help boost performance on captioning metrics. The proposed method can be a useful trick in captioning network implementations for small performance gains. Post-rebuttal comments -- The paper proposes a modification that boosts performance in practice. The author feedback clarifies my concerns regarding experimental setup of the result tables and with this additional explanation, the tables are consistent. I still feel experimentation on different spatial features (the core idea of the method) is missing, the attention visualizations are interesting and should be discussed more in the final version.

Reviewer 3



Overall I liked the paper. It is well written and the task is interesting. I have some minor concerns regarding novelty at both the conceptual and technical levels. First, the reference supporting the claim that "incorporating spatial relationships has been shown to improve the performance of object detection" shows a lack of knowledge of important works in computer vision (prior to the deep learning era). For example, Hoiem et al.'s "Putting Objects in Perspective", ECCV 2006 showed the same thing over 10 years ago. Second, the object relation transformer model is a small variation of the standard transformer. I congratulate the authors on running and reporting statistical tests on their experimental results. They also provide sufficient implementation details to reproduce the models. Providing code on acceptance would help reproduceability further, and the authors should consider doing so. Some technical/minor comments: 1. In Section 3.2 it is not clear which parameters are shared between different heads and which are unique to each head. The authors should clarify and index the head-specific parameters. 2. In Eqn (6) the normalization in the second log term should be h_m. Moreover, what happens if the objects are aligned horizontally or vertically, i.e., x_m - x_n = 0? 3. L155 "\Sigma_A = with \Sigma" -> "\Sigma_A with \Sigma"

[Author Response · NeurIPS 2019]

We thank the reviewers for the time and attention invested in these reviews. We address the reviewers' remarks below.

**> Include discussion of [A,B,C,D] and empirical comparison to [A,B,C, 21]**

We will add empirical comparison with [A,B,C] (the authors of [21] have not released their model, nor did they present evaluation on the MSCOCO dataset). The following are the conceptual comparisons to be added to Related Work: [A] does not consider the size and position of detected regions in their algorithm. [B] classifies the spatial relationships of two boxes into 11 classes, such as "inside", "cover" or "overlap". Our approach directly utilizes the size ratio and difference of x- and y-location to compute the box relationship, which implicitly encodes the mentioned properties. [C] does incorporate object spatial relationships, but does not use visual geometry to do so. [D] does successfully learn relationships between all object pairs of objects; however, it is not comparable to the image captioning task as it does not provide a single caption describing the overall image scene. In addition, all of these works are using LSTMs while our algorithm uses the transformer architecture for the caption generation.

**> Include evaluation on the online MSCOCO test server.**

The MSCOCO test server results from our model are shown below. Note that the model used in this submission was trained on the Karpathy train split, rather than the standard practice of training it on Train+Val. For a fairer comparison, we will do so in the camera-ready paper version.

| test set | CIDEr-D | BLEU-1 | BLEU-2 | BLEU-3 | BLEU-4 | METEOR | ROUGE-L |
|---|---|---|---|---|---|---|---|
| c5 / c40 | 123.6 / 125.4 | 80.2 / 94.1 | 64.7 / 88.3 | 50.0 / 79.3 | 37.8 / 68.7 | 28.5 / 37.4 | 58.1 / 72.9 |

**> If time permits, include a human evaluation, comparing the proposed model to strong baselines or SoTA models.**

A comparison based on human judgments should give additional insights, and we'll add it to the camera-ready paper.

**> Discuss failure modes of the proposed method.**

We will analyze and discuss failure modes along with illustrative examples, space permitting. We manually reviewed results from our best model over 100 randomly sampled images from the MSCOCO test set (and will extend the analysis to the complete test set for the camera-ready submission). Out of the 62 observed errors, 58% pertained to objects or things (24% missed, 24% wrong, 10% extraneous), 21% to relations (5% missed, 16% wrong), 16% to attributes (5% missed, 8% wrong counts, 1.5% wrong colors, 1.5% extraneous), and 5% to syntactic errors.

**> Do all methods in Table 1 use the same beam size? If not, what fraction of the improvement is attributed to performing beam search? The authors should compare the methods with same beam size, and clarify it on Tab. 1**

Our best results were obtained with beam size 2, in consistency with other research on Image Captioning optimization [20] (appendix A). Only in Tab. 1, for a fair comparison with other models in the literature, we present our result with the same beam size of 5 that they used to communicate their performance, indicating it in the table's caption. We will make this distinction more clear in the Implementation Details Section, by specifying the setting for each experiment. Finally, the last line of our ablation study in Tab. 3 addresses the fraction of the improvement attributed to beam search.

**> There is a difference in the scores for the Up-Down baseline between Tab. 3 and Tab. 1. Is it because self-critical training was not used for Tab. 3? Are the splits different?**

That is correct, self-critical training was not used in Tab. 3. In addition, we used our own implementation of the Up-Down algorithm for the baseline results in Tab. 3. The splits are the same in all the tables.

**> For the proposed features, the improvement on "size" sub-score of SPICE is not statistically significant. In such a scenario, it would be help to study different relative location features for the two bounding boxes.**

We experimented with different object geometric features, as well as different ways of fusing these with the object appearance features. We did not try the suggested technique from Interaction Pattern of Chao et al. in "Learning to detect human-object interactions", which does encode the relative size of pairs of objects. Since this method scales the bounding box 0-1 masks of pairs of objects, it may not address the issue identified by the SPICE absolute object size metric. But we are enthusiastic to check if we can get a boost following the suggested technique.

**>A visualization for the attention weights for an appropriate layer (say for the qualitative examples in Tab. 7) may be useful to demonstrate that the network does indeed learn spatial relationship information.**

We generated the following visualization of the self-attention in our proposed Object Relation Transformer, here displayed for one of the relation example images (averaged across attention heads, and self-attention layers). The detected object transparency is proportional to the attention weight with respect to the chair outlined in red. This image will be discussed in the paper.

**Generated Caption**: *two beach chairs under an umbrella on the beach*

**> What happens on other vision and language tasks?**

We are considering applying this approach to other tasks (HOI and VQA).

[Meta-Review · NeurIPS 2019]

An object relation module is included into the transformer model. Improvements are demonstrated using this approach. After reading the rebuttal the reviewers agreed that this is an interesting direction to pursue. The reviewers liked the method and partly the results presented in the rebuttal. However the reviewers also remained concerned that additional evidence is necessary (e.g., proper evaluation on test server, experimentation with different spatial features, more in-depth discussion of the attention visualizations, empirical comparison to prior work and human evaluation). AC concurs and recommends acceptance as a poster.